# Wind as a natural hazard in Poland

Tadeusz Chmielewski[1], Piotr A. Bońkowski[1]

[1]Faculty of Civil Engineering and Architecture, Opole University of Technology, Prószkowska 76, 45-758 Opole, Poland

*Correspondence to*: Tadeusz Chmielewski (t.chmielewski@po.edu.pl)

**Abstract.** The paper deals with wind speeds of extreme wind events in Poland and the descriptions of their effects. Two recent estimations developed by the Institute of Meteorology and Water Management in Warsaw and by Professor Lorenc (a previous co-worker of the Institute) are presented here. Their strong and weak points are briefly described. The 37 annual maximum gusts of wind speeds measured at all meteorological stations between 1971 and 2007 are analysed by an extremal probability paper, block-maxima (BM), and peak-over-threshold (OVT) approaches. Based on the measured and estimated wind speeds
(taken from existing literature), the authors suggest new estimations for extreme winds that may occur in Poland. Shortly, Poland will construct important structures, such as a central air terminal and (some) nuclear power plants, so knowledge about extreme winds in our country is significant for engineers who will design these structures.

## 1 Introduction

The wind is the dominant environmental load affecting structural design in the world (e.g. Krishna, 1995; Goliger et al., 2013).
In many countries, as well as in Poland, it is responsible for damage to ecosystems (Pawlik and Harrison, 2022; Pettit et al., 2021) and structures (Unanwa et al., 2000; Chmielewski et al., 2020). Moreover, hurricanes and tornadoes can cause economic losses and have a negative impact on society (Fricker, 2020; Kafi et al., 2021). Also, in Poland, wind storms such as synoptics, thunderstorms, and downslope winds in the Tatry and Karkonosze mountain regions, tornadoes, downbursts, and derechos are the leading causes of economic loss (see e.g. damage model for Europe by Koks and Haer., 2020).
From the engineering perspective, it is important to consider rare and extreme events to design a critical infrastructure. Properly estimating these extreme events and analysing their effects on structures and infrastructure are still current research issues. Examples of such analyses for the European region can be found in the paper by Outten and Sobolowski (2021), who used Euro-CORDEX regional climate models, in the paper by Della-Marta et al. (2009), who used ERA-40 reanalysis data, or in the paper by Taszarek et al. (2020a, b), who used e.g. European Severe Weather Database (ESWD) for their analyses.
The recent estimation of the wind speed for all types of wind events was developed by Lorenc (2012). Five names were proposed for gust wind speeds ranging from 11 to 32 m/s at 10 m above ground: violent wind, storm, strong wind, and hurricane wind. It is not used in everyday life, and this paper's authors oppose this proposal (for example, no tropical storms occur in Poland, i.e., no hurricanes). The authors' study of maximum wind speeds in Poland is based on a set of annual maximum gust wind speeds measured at 39 meteorological stations from 1971 to 2010 (40 years), tornado reports collected from 1899 to 2019

(120 years), and estimation of wind speeds of derechos and recent tornadoes in Poland (Polish Government Centre for Security, 2013; Chmielewski et al., 2013; Taszarek and Brooks, 2015; Chmielewski et al., 2020). Based on the available data, the authors' proposals for maximum wind speeds in Poland are given in Tables 2 and 3.

The case study of estimating intense and extreme wind speeds in Poland based on the measured and estimated wind speeds is analysed in Section 3. In Section 4, a new radical wind estimation is proposed. Shortly, Poland will construct some critical
structures, such as a central air terminal and (some) nuclear power plants.

## 2 Review of existing wind types and classifications in Poland

Poland is located in the centre of Europe between the Baltic Sea in the north and the mountain ranges of the Carpathians and Sudetes in the south. So, the two types of climatic/conditions prevail, i.e. the continental and coastal. The country is generally flat and situated two or three hundred meters above sea level. Five different types of wind events may be identified in Poland.
They are western or northern–western extratropical cyclones; downslope winds in the Tatra and Karkonosze mountain regions (foehn winds); thunderstorms; tornadoes which belong to the most devastating wind events, as described in (Chmielewski et al., 2013); derechos, an example of such event is described in the paper (Chmielewski et al., 2020).

Two classifications of weak and strong winds exist in Poland. The first was performed by the Institute of Meteorology and Water Management (IMWM) (only for weak and strong winds, not for extreme winds such as tornadoes or derechos) (Polish
Government Centre for Security, 2013; Highlighted threat thresholds | IMWM, 2023). Three threats were considered for three different wind speeds with a description of the effects of the wind action as follows: 1st-degree threat: $V_{av} > 15$ m/s or $V > 20$ m/s, damage to buildings, roofs, damage to trees, breaking branches and trees, traffic difficulties, 2nd-degree threat: $V_{av} > 20$ m/s or $V > 25$ m/s, damage to buildings, roofs, breaking and uprooting trees, difficulties in communication, damage to overhead lines; 3rd-degree threat: $V_{av} > 25$ m/s or $V > 35$ m/s, destroying buildings, tearing off roofs, destroying overhead lines, large
damage to trees, significant difficulties in communication, and being life-threatening.

The second classification was proposed by Lorenc (2012), a co-worker at the Institute of Meteorology and Water Management. This proposal comprises eight classes in the range of wind speeds from 11 to more than 251 m/s, with a damage description for each class. These classes were as follows: gusty wind, violent wind, storm, strong wind, hurricane wind, hurricane/tornado 1st degree, very strong hurricane/tornado 2nd degree, and destructive hurricane/tornado 3rd degree. As the introduction
mentions, this proposal is not used in everyday life. The authors of this study are opposed to this proposal. We do not have tropical storms in Poland; therefore, the name hurricane should not be used.

## 3 Materials and Methods
### 3.1 Yearly maximum wind speeds measured at the meteorological stations in Poland
To analyse the return period of extreme winds speeds, wind speed records in gusts measured in 1971-2010+ were used. These extreme winds were in the range of 25 – 40 m/s, with extreme values recorded at: 6.11.1986 Bielsko Biała 48 m/s, 1.12.1975

Zakopane 47 m/s, 21.10.1986 Kalisz 46 m/s, 8.02.1990, Łeba 44 m/s, 4.12.1999 Hel 41 m/s. The sample data set of annual maximum wind speeds in m/s is as follows [25, 26, 26, 27, 27, 29, 29, 30, 30, 31, 31, 32, 32, 33, 34, 34, 34, 34, 35, 36, 36, 37, 37, 37, 37, 38, 38, 38, 39, 39, 40, 41, 43, 44, 46, 47, 48].

65 For the above data set, three approaches have been applied to identify and model the extrema: The Gumbel Extremal Probability Paper, the block-maxima approach where the extrema follows a generalised extreme value distribution (BM-GEV), and the peak-over-threshold that fits the extrema in generalised Pareto distribution (POT-GPD).

### 3.2 The Extremal Probability Paper

In Fig. 1, 37 annual maximum gust wind speeds measured at all meteorological stations during 1971–2007 (Lorenc, 2012) are
70 plotted on the Gumbel probability paper. The red line indicates the regression line fitted to the data with the following coefficients: $u_n = 32.4$ and $1/a_n = 5.14$. The regression line fits the measured data well, meaning that the random phenomenon of extremal wind speed described at this point can be modelled with a log-normal distribution, also known as the Gumbel distribution, an extreme-value distribution.

From the straight-line graph, the probability associated with annual wind speeds in Poland for a given value can be read
75 directly. For example, the annual probabilities of gust wind speeds exceeding magnitudes of 30 and 40 are as follows:
$P(v_n > 30) = 1 - F_{V_n}(30) = 1 – 0,21 = 0,79$; $P(v_n > 40) = 1 - F_{V_n}(40) = 1 – 0,80 = 0,20$. The return period for wind speeds greater than 45 m/s can be easily estimated. For example, the 50 years return period corresponds to $s = -\log(-\log(0.98)) = 3.90$ and wind speed $V_n = 32.4 + 5.14 \cdot 3.90 = 52.4$ m/s. The authors would like to remark that the analysis is based on historical data. However, some studies predict an increased frequency of severe wind occurrence (Rädler et al., 2019) and changes in the
80 gust field due to climate change (Schwierz et al., 2010), which may influence current return period predictions.

### 3.3 The Block Maxima Method

The Block Maxima Method analysis has been done using the following steps. After the division of our dataset into non-overlapping blocks of 6 elements (for the two first blocks) or five elements (for the rest blocks), we obtained a new block
85 maxima as follows [39, 47, 48, 44, 36, 38, 43]. The statistical software Pythons scipy.stats was used in the inverse cumulative distribution function (genextreme.ppf function) to fit the Generalized Extreme Value. It resulted in the following distribution parameters: Shape ($\xi$): 1.1276862002991992; Location ($\mu$): 42.75887436787345; Scale ($\sigma$): 5.910345049383524. With this method, the following wind speeds were obtained for 10-, 20- and 50-year return periods:
10-year return level: 47.59 m/s; 20-year return level: 47.82 m/s; 50-year return level: 47.94 m/s.

90

### 3.4 The Peak-Over Threshold Method

The Peak-Over Threshold Method analysis has been done using the following steps. In the first step, we must choose a threshold above which we consider data points as extreme events. Let the threshold = 45 m/s. This method uses values that

exceed a certain threshold to define extreme events. This threshold was set at 45 m/s, which left us with three values that
exceeded it, presented in the table below:

**Table. 1. Wind events above the selected threshold**

| Year | Wind speed[m/s] |
|------|-----------------|
| 1976 | 47          100 |
| 1986 | 46              |
| 1989 | 48              |

Then, following the assumption that the exceedances follow the generalised Pareto distribution (GPD), these values were used
to determine the parameters of Pareto distribution by applying the genpareto.fit function of the scipy.stats Python library. With
these parameters established, the inverse cumulative distribution function (genpareto.ppf function) was used to make long-
term predictions and create a distribution plot (Fig. 2). It resulted in the following distribution parameters: Shape Parameter
($\xi$): -2.372618164968692; Location Parameter ($\mu$): -0.6502177076262381; Scale Parameter ($\sigma$): 115.42839026279556. The
following wind speeds were obtained for 10-, 20- and 50-year return periods: 10-year Return Level: 47.79 m/s; 20-year Return
Level: 47.96 m/s; 50-year Return Level: 48.00 m/s.

Comparing these three approaches gives similar results for a 50-year return period of around 50 m/s. From the analysis, it is
predicted that 10-, 20-, and 50-year wind yield a similar threat of P2 (Table 3), according to the proposed new classification
described in the next section.

**3.5 Estimated wind speeds of tornadoes and derechos**

To further analyse possible extreme wind speeds in Poland, an estimation of wind speeds using observation of damage caused
by tornadoes and derechos can be used. Taszarek and Brooks (2015) described the updated climatology of tornadoes in Poland
and the significant problems related to the database. A total of 269 tornado cases derived from European Severe Weather Data
were used in the analysis, and the tornadoes were divided according to their strengths. On average, 8–14 tornadoes (including
2-3 waterspouts) with two strong tornadoes occur yearly, and one violent tornado occurs every 12–19 years. Cases of strong
and even violent tornadoes that cause death indicate that the possibility of a large fatality tornado in Poland cannot be ignored.
The estimated extreme wind speeds of tornadoes and derechos were based on recent studies of two case studies of these
phenomena, which have occurred in Poland (e.g. Chmielewski et al., 2013, 2020). In the first case, i.e., on the 15th of August,
2008, a tornado caused severe damage in the three provinces of Poland: Opole, Katowice, and Łódź. Along the 105 km path,
1624 buildings were damaged, 4 people were killed, 60 people were injured, and some livestock were killed. Two approaches
were used to estimate the tornado's wind speed. The first one was based on a comparison of the examined damage caused by

the tornado in the affected area with the TORRO (tornado intensity scale). The second approach is based on the structural analysis of the destroyed freestanding structures. Three road signs, which were bent while the tornado passed, were examined in these studies. In the first approach, wind speeds were estimated in the range of 52–72 m/s at a reference height of 10 m. In the second approach, the wind speed was approximately 71 m/s at 2.3 m above the ground.

In the second case, a strong thunderstorm occurred on August 11–12, 2017, resulting in catastrophic damage in three provinces in Poland: Wielkopolskie, Kujawsko-Pomorskie, and Pomorskie. This disaster resulted in the deaths of six people, injuries of several dozen others, and enormous property losses. The event has been described and analysed in a previous study (Chmielewski et al., 2020). A house with its roof blown off is an excellent example of wind speed for this derecho. The weight of the roof was calculated based on the rafter framing of the house. By estimating the connection strength between the rafter

plates and knee walls, it was possible to calculate the total force required to blow off the roof of the house. Subsequently, the pressure coefficients were obtained from the Tokyo Polytechnic University aerodynamic database. The aerodynamic force acting on the blown-off roof was calculated for the low-rise building with a gable roof because of the similar ratios of length, width, and height. By comparing the aerodynamic force with the total force required to blow off the roof of the house, it was possible to calculate the critical wind speed necessary for roof blow-off. The critical wind speed is approximately 60 m/s. It

was much larger than wind speeds measured by meteorological stations on the path of the derecho in Chojnice (31,2 m/s), Gniezno (34,8 m/s), Chrząstowo (36 m/s), and Elbląg (42 m/s).

It is important to note that for both events analysed here, the estimated wind speeds are also higher than extreme wind speeds recorded by meteorological stations in the period 1971-2010+. It means that wind speeds predicted for respective return periods using methods presented in sections 3.2-3.4 may be underestimated.


## 4 Development of estimation of maximum wind speeds in Poland

As indicated in Section 2, two wind speed classifications exist in Poland. Both of them have some drawbacks, i.e. in the classification used by IMWM (Highlighted threat thresholds | IMWM, 2023), the description of wind effects is not detailed

enough, and the scale ends at the gust wind speed equal to 32 m/s, while Lorenc's (Lorenc, 2012) classification considers e.g. hurricanes that do not occur in Poland. For these reasons, the authors suggest a new estimation for strong and extreme winds in Poland based on the observation of their effects.

It is assumed that the border between strong and extreme wind is the wind speed of 30 m/s. For the new estimation, the author proposes Tables 2 and 3. To create Tables 2 and 3, the estimation developed by the IMWM, Lorenc, Fujita (F-Scale), and

Mehta (EF-Scale) were considered. In Table 2, the first two columns are the same as by the IMWM (with $V \leq 30$ m/s for the 3rd degree). However, the description of the effects of wind action is based on ten years of the author's observation of wind damage for particular degree threats after wind events. To consider higher wind speeds, Table 3 for extreme winds is proposed (P-scale). It is the modification of the EF-scale for tornado intensity developed in the USA for Tornado Intensity (Mehta, 2013). The modification was needed to match wind speeds from Table 2 (especially for EF0 and EF1, into P0 and P1 degrees).

**Table 2. Classification of the levels of threats caused by strong wind [IMWM + Authors]**

| Degree threats | Wind speed criteria [m/s] | Description of the effects of wind action |
|---|---|---|
| 1 | $V_{av} > 15$ or $V > 20$ | It moves tree branches, billboards, and road signs. Some of them can be knocked down. It breaks weaker tree branches that can block communication routes. It tears off individual roof tiles, scatters garden furniture, and damages local power lines, tents and awnings. Vehicle drivers feel wind speed. Light objects float in the air. During snowfall, the wind causes blizzards. |
| 2 | $V_{av} > 20$ or $V > 25$ | It breaks tree limbs, breaks or tears up shallow-rooted trees. Broken branches and trees block roads, trams, and railway lines. The tree branches can fall on vehicles. Broken power cables (tens of thousands of people are deprived of electricity). Wind significantly damages roofing, old farms, and residential buildings. During wind gusts, cars are pushed to the sides of the road. It can overturn billboards, road signs, and fences; individual items float in the air after an intense storm; and cellars and apartments are flooded. |
| 3 | $V_{av} > 25$ or $V > 30$ | Such synoptic wind speeds rarely occur in Poland - once every few years, their effects are similar to those described in the case of hazard level 2, but in larger dimensions, e.g., there are significantly damaged or completely broken roofs, damaged farm buildings and residential buildings, broken power poles and cables, roofs with reinforced concrete and steel structures are damaged. The described nature of the damage is typical for weaker tornadoes and squalls. |

$V_{av}$ – mean wind speed, V- gust wind speed

**Table 3. Classification of extreme wind speeds. Comparison of EF-scale (Mehta, 2013) and proposal of P-scale by authors.**

| EF0 | EF1 | EF2 | EF3 | EF4 | EF5 | Wind speed |
|---|---|---|---|---|---|---|
| 105-137 | 138-178 | 179-218 | 219-266 | 267-322 | >322 | km/h |
| 29.2-38.1 | 38.6-49.4 | 49.7-60.6 | 60.8-73.9 | 74.2-89.4 | >89.4 | m/s |
| | **P1** | **P2** | **P3** | **P4** | **P5** | |
| | **121-170** | **171-220** | **221-270** | **271-324** | **>324** | **km/h** |
| | **33.6-47.2** | **47.5-61.1** | **61.4-75** | **75-90** | **>90** | **m/s** |

## 5 Conclusions

The authors formulated the following conclusions based on the analyses of measured wind speeds and annual maximum wind speeds in Poland in the past and damage observations caused by strong and extreme winds:

a) The proposed wind speed estimation has been developed using data available in Poland, as described in Section 3. It estimates wind speed for strong winds (Table 2) and extreme winds (Table 3) as tornadoes, derechos, etc.

b) The 37 annual maximum gusts of wind speeds measured at all meteorological stations between 1971 and 2007 are

analysed by the extremal probability paper, block-maxima (BM), and peak-over-threshold (OVT) methods. The results of these analyses are similar. The 50-year Return Period for these three methods is around 50 m/s.

c) Two recent extreme wind events in Poland and their wind speeds are summarised. The wind speeds estimated by observing damage caused by them are higher than wind speeds recorded by meteorological stations.

d) The authors propose for synoptic, downslope winds in the Tatra and Karkonosze mountain regions, the estimation of

strong winds proposed by IMWM with an improved description of the effects of wind action (Table 2), and adaptation of the EF tornado intensity scale with some modifications to extreme winds presented in Table 3 (the P-scale).

As the wind is one of the most important loads during structural design, properly estimating wind speed return periods is crucial. In this paper, wind speeds are estimated using recorded data and three methods are compared. Recorded wind speeds, however, are lower than the ones estimated from the observed damage of two extreme events. It indicates that apart from

predicted changes in future wind speeds, the wind speeds for respective return periods estimated using only recorded data may be underestimated. This may be important regarding the design of the most important structures in Poland. For this reason, it may be beneficial to include data from the observed damages in future wind speed predictions.

### Code and data availability

Data and codes can be made available upon request to the corresponding author.

### Author contribution

TCh: Conceptualization, Formal analysis, Investigation, Methodology, Visualisation, Writing – original draft preparation, Writing – review & editing. PB: Formal analysis, Visualisation, Writing – review and editing.

### Competing interests

The authors declare that they have no conflict of interest.

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

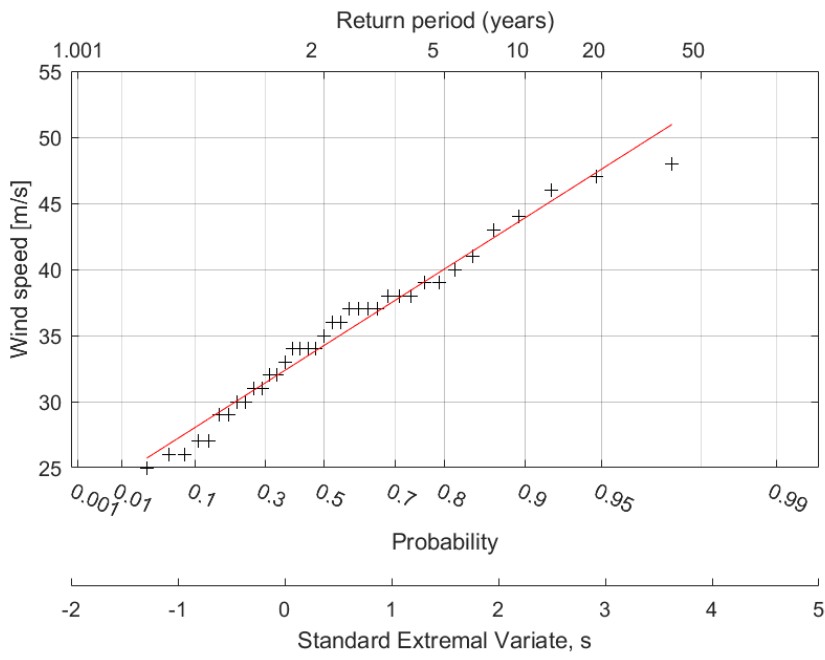

240

**Fig. 1. Plot of wind speeds from (Lorenc, 2012) against standard extremal variate *s*.**

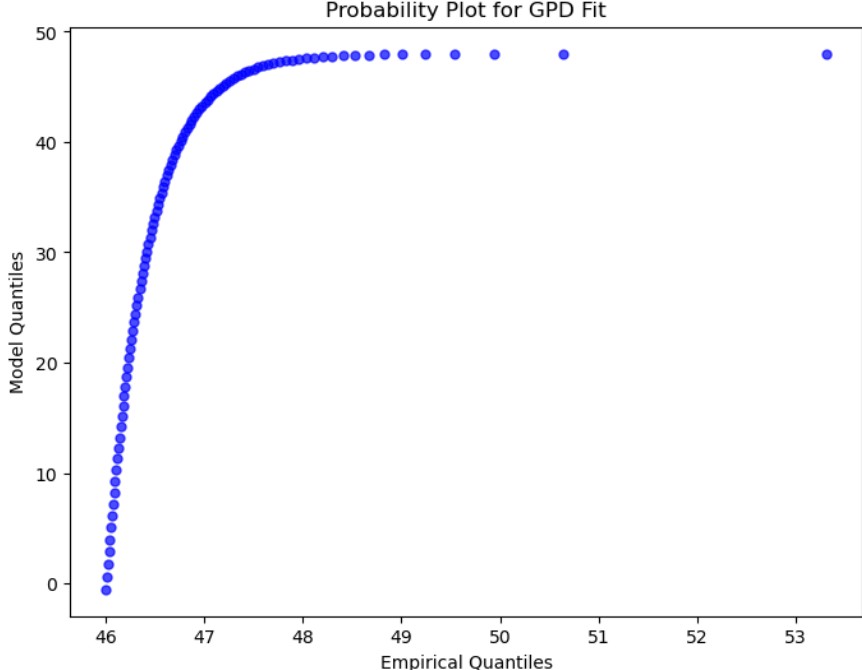

**Fig. 2. Distribution plot obtained with The Peak-Over Threshold Method.**