# Peer review of "Wind as a natural hazard in Poland"

_EGUsphere, 2023_

## Community Comment (CC1)

In general, through the study of historical documents and weather station data, combined with existing wind speed classification standards, the author uses the extreme value probability method to put forward targeted wind speed classification standards suitable for Poland. This paper is innovative to a certain extent, but there are not enough cases to support the proposed wind speed classification standards. There is also a lack of further research to improve the accuracy of its wind speed classification, and the accuracy of its wind speed classification lacks convincing theoretical support.

**Technical Corrections:**

1. In paragraph 35, "the two types of climatic prevail" should be specified

2. In paragraph 60, "4.2 The Extreme Probability Paper," can the author find wind speed data recorded by meteorological stations over the past decade to model? After all, the climate over the past decade has undergone significant changes compared to the early 21st century, and using data from recent years is more conducive to enhancing persuasiveness.

3. In paragraph 80-85, how is the wind speed obtained by the first method specifically determined.

4. In the final Tables 1 and 2, the application should be based on the author's classification criteria, which can be presented as a small example, so that the classification criteria have practical application value.

---

## Author Comment (AC1)

Thank you for the statement that the subject matter is interesting from a scientific point of view and the approach is correct. We are sorry that for you is incomplete and messy. But you should remember the note: **writing a major academic work is difficult enough, but is even more so when working in a foreign language, it is easy to make some mistakes.**

**It is true that references [3], and [4] in the abstract are unexplained.** The reference[4] is explained in Section 3 and enlarged in Table 1 (see [IMWM+ Authors]). The reference [3] is shortly described also in Section 3 (raws 49-53).

**(uppercase and lowercase letters like r.31 "Section")** is a rather minor mistake, which might be corrected by the editorial office. It is a pity that you as the reviewer not suggested us some corrections to our English ( if English is your native language).

**-section 4: understandable but written in a very concise way, with the data and calculations reported in the text creating confusion**.

Yes, it is true that this subsection is written in a concise way. This was our purpose to write this paper as short as possible. Accordingly, if N largest values from a population of the exponential type are plotted on the extremal probability paper, such that the mth value (in increasing order) is plotted at the probability m/(N+1), the results should show a linear trend with a positive slope. In Fig. 1. a linear trend is observed. This means that the random phenomenon described in this point can be modeled with a log-normal distribution, known also as the Gumbel distribution, one of the extreme-value distributions. From the straight line graph, the probability associated with annual wind speeds in Poland of a given value may be read off directly as follows

P(vn > given value) = 1 - FVn (given value) = 1 – probability from the horizontal axis. For example, the annual probability of gust wind speeds exceeding magnitudes 40m/s is

P(vn > 40) = 1 - FVn (40) = 1 – 0.80 = 0.20.

**-section 5: how were Tables 1 and 2 produced?**

The author suggests a new estimation for strong and extreme winds that may occur in Poland in the future based on old and recent works [1, 2, 3, 4, 5, 6, 7] and the EF-Scale for tornado Intensity [8]. Table 1 is a modification of the proposal of the IMWM. Due to several years of

authors' observation, we prepared a description of the effects of wind action for each degree of threat. Table 2 also modifies the EF scale, especially for degrees EF1 and EF2. It is assumed that the border between strong and extreme winds is the wind speed of 108 – 120 km/h. The description of the effect of extreme winds in the P scale is the same as the EF scale.

**-section 6: does it make sense to insert a section for the reported content?** Yes, it makes sense for engineers who will deal with the design of these new future structures in Poland. The ISO 1382 standard can help them to overcome these difficulties.

Last author's remark.

We are very grateful to the unknown referee for his detailed and fruitful review. Thank you very much for your effort.

---

## Author Response (AR1)

**Response to Reviewers' Comments**

We would like to sincerely thank the editor and reviewers for their valuable comments that helped significantly improve the manuscript. Below response, and corrections to the comments are given:

| Rev. | Comm. | Reviewer comment | Response | Location in the corrected manuscript |
|---|---|---|---|---|
| E | 1 | Particularly the review of the state of the art needs to be improved. There are few references - about 20 are recommended - to prove that there is a discussion on the matter. | We have improved the literature review by adding more references | Introduction section, Lines 81-82 |
| | 2 | Also, and especially related to the short list of references, there are many self-citations. | Two self-citations was removed. | All text |
| | 3 | The discussion part also needs to be extended in order to proper support the conclusions. This applies to comments on the figures and tables in particular. | New analyses are now included with additional discussion regarding the 50-year wind speed, which is the most important for the design of new structures. | Section 4 |
| 1 | 1 | Bibliography almost absent...Two references ([3], [4]) appear in the abstract that are unexplained; | We have improved the literature review by adding more references. The references [3] and [4] were removed from the text | Abstract, Introduction section, Lines 81-82 |
| | 2 | -Little attention to details (uppercase and lowercase letters like r.31 "Section"), as well as a sometimes inaccurate use of English (i.e. r.102 "Strong and intense winds are the two types of winds into which it is classified") and a level of writing that is sometimes inappropriate for a scientific article (r.53 "And the authors of this paper are opposed to this proposal. We do not have tropical storms in Poland, so the name hurricane should be not used"); | The manuscript is now proof-read | All text |
| | 3 | section 4: understandable but written in a very concise way, with the data and calculations reported in the text creating confusion; | Section 4 is now rewritten including data used for the analyses, and calculations using BM and POT methods | Section 4 |
| | 4 | section 5: how were tables 1 and 2 produced? the section should contain the innovative and publication- | To create Table 1 and Table 2 the estimation developed by the IMWM, by Lorenc, by Fujita | Lines 151-159 |

| | | | | |
|---|---|---|---|---|
| | | worthy aspect of the work, which is missing or not highlighted by the authors. Furthermore, the tables are reported with the wrong numbering in the text. | (F-Scale), and by Mehta (EF - Scale) were taken into account. Additionally, the author's observation of the wind damage was considered. Lines 151-159 have been rewritten to clarify this matter. The numbering is now corrected | |
| | 5 | section 6: does it make sense to insert a section for the reported content? The idea of inserting the procedure is good, but how are the contents of the paper and ISO 1382 connected? Nothing is explained...Furthermore, the idea included in the abstract on the future construction of structures in Poland, for which studies of this type are needed, is also interesting, but it is not at all detailed in the paper. | Section 6 has been now removed | - |
| 2 | 1 | you should use block maxima (BM) or peak-over-threshold (POT) method for extreme winds | The analyses using BM and POT methods are now included in the paper | Section 4.3 and 4.4 |

---

## Author Response (AR2)

**Response to Editor's Comments**

We would like to sincerely thank the editor for valuable comments that helped us improving the manuscript even further.

We have corrected section headings. In section 5 (section 4 in corrected manuscript) we primarily aimed to improve wind classification currently used in Poland. This matter is now clarified in the manuscript. We extended conclusions and included discussion part. Additionally, reference list was slightly extended. There are numerous studies on the wind and its effect to the structures and society but we did not want to overextend introduction part.